# Menstrual hygiene practice among adolescent girls in Ethiopia: A systematic review and meta-analysis

Biniyam Sahiledengle[1]*, Daniel Atlaw[2], Abera Kumie[3], Yohannes Tekalegn[1], Demelash Woldeyohannes[4], Kingsley Emwinyore Agho[5]

1 Department of Public Health, Madda Walabu University Goba Referral Hospital, Bale-Goba, Ethiopia, 2 Department of Human Anatomy, Goba Referral Hospital, School of Health Science, Madda Walabu University, Bale-Goba, Ethiopia, 3 Department of Community Health, College of Health Science, Addis Ababa University, Addis Ababa, Ethiopia, 4 Department of Public Health, College of Medicine and Health Science, Wachemo University, Hossana, Ethiopia, 5 School of Health Sciences, Western Sydney University, Penrith, NSW, Australia

* biniyam.sahiledengle@gmail.com

## Abstract

### Background

Adolescent girls face several challenges relating to menstruation and its proper management. Lack of adequate sanitary products, inadequate water supply, and privacy for changing sanitary pads continue to leave adolescent girls with limited options for safe and proper menstrual hygiene in many low-income settings, including Ethiopia. These situations are also compounded by societal myths, stigmas surrounding menstruation, and discriminatory social norms. This systematic review and meta-analysis aimed to estimate the pooled proportion of safe menstrual hygiene management among adolescent girls in Ethiopia using the available studies.

### Methods

We searched PubMed, Google Scholar, African Journal Online (AJOL), Hinari, Science Direct, ProQuest, Direct of Open Access Journals, POPLINE, and Cochrane Library database inception to May 31, 2021. Studies reporting the proportion of menstrual hygiene management among adolescent girls in Ethiopia were considered. The Cochrane Q test statistics and $I^2$ tests were used to assess the heterogeneity of the included studies. Since the included studies revealed considerable heterogeneity, a random effect meta-analysis model was used to estimate the pooled proportion of menstrual hygiene management (MHM).

### Results

Of 1,045 identified articles, 22 studies were eligible for analysis (n = 12,330 participants). The pooled proportion (PP) of safe MHM in Ethiopia was 52.69% (95%CI: 44.16, 61.22). The use of commercial menstrual absorbents was common 64.63% (95%CI: 55.32, 73.93, $I^2$ 99.2%) followed by homemade cloth 53.03% (95%CI: 22.29, 83.77, $I^2$ 99.2%). Disposal of

**Data Availability Statement:** All relevant data are within the paper and its Supporting Information files.

**Funding:** The author(s) received no specific funding for this work.

**Competing interests:** The authors have declared that no competing interests exist.

absorbent material into the latrine was the most common practice in Ethiopia 62.18% (95% CI: 52.87, 71.49, $I^2$ 98.7%). One in four girls reported missing one or more school days during menstruation (PP: 32.03%, 95%CI: 22.65%, 41.40%, $I^2$ 98.2%).

## Conclusion

This study revealed that only half of the adolescent girls in Ethiopia had safe MHM practices. To ensure that girls in Ethiopia can manage menstruation hygienically and with dignity, strong gender-specific water, sanitation, and hygiene (WASH) facilities along with strong awareness creation activities at every level are needed.

## Introduction

Menstrual hygiene management (MHM) is a vital part of the health and dignity of women and girls worldwide. Menstrual Hygiene Management (MHM) refers to '*Women and adolescent girls using a clean menstrual management material to absorb or collect blood that can be changed in privacy as often as necessary for the duration of the menstruation period, using soap and water for washing the body as required, and having access to facilities to dispose of used menstrual management materials*' [1, 2]. Globally, at least 500 million women and girls lack adequate facilities for menstrual hygiene management. Lack of WASH (water, sanitation, and hygiene) facilities, particularly in public places, such as in schools and workplaces, can pose a major obstacle to women's and girl's menstrual hygiene [3]. Also, societal myths and stigmas surrounding menstruation, discriminatory social norms, cultural taboos, poverty, and inadequate basic services often cause adolescent girl's menstrual hygiene needs to go unmet [1].

Several factors affect adolescent MHM experiences, including inadequate WASH facilities and lack of separate toilets with doors that can be safely closed in schools, unavailability of means to dispose of used sanitary pads and water to wash hands, lack of absorbent materials, fear of blood leaking, menstrual pain, and inadequate knowledge towards the menstrual cycle and its management were key challenges that adolescent girls face in maintaining their menstrual hygiene in a private, safe and dignified manner [3–5].

In developing countries, the problem of menstrual hygiene gets worse, as adolescent girls can face significant challenges in managing their menstruation, particularly when attending school [6]. Girls' inability to manage their menstrual hygiene in schools results in school absenteeism, resulting in poor school performance, drop-out, and reduced educational attainment [3, 6–8]. In some instances, girls were obliged to carry soiled absorbents back home in schools that did not have waste disposal facilities and separate toilet facilities for female students [9]. A growing body of evidence shows that considerable school absenteeism due to menstruation among adolescent girls was a frequent phenomenon in many low and middle-income countries (LMICs) [4, 5, 7, 9, 10]. Besides, lack of clean, safe, functional, private, and gender-specific WASH facilities in school settings and poor access to sanitary materials were also associated with poor MHM among adolescent girls [4, 8].

Studies further point out that, absence of affordable menstrual absorbent materials makes adolescent girls use inappropriate and unhygienic absorbent materials such as newspaper, old rags, underwear, homemade clothes, sponges, dried leaves, or socks to collect menstrual blood and manage their periods [7, 10–13]. Unhygienic MHM predisposes adolescent girls to reproductive tract infections with potential long-term effects on their reproductive health [14]. In recent years, a growing body of studies demonstrates a strong and consistent association between poor menstrual hygiene practices and a higher prevalence of lower reproductive tract

infections (RTI) [15–17]. It is suggested that, during menstruation, girls need to change their sanitary absorbent regularly and menstrual absorbent materials should be changed three or four times a day, and girls need to wash their genital area at least twice a day with soap and water [10, 18].

In Ethiopia, most school adolescents practiced unsafe menstrual hygiene during menstruation [7, 19–21], and only 1 in every 4 girls knew about menstruation before their first period [22]. Moreover, studies from Ethiopia revealed that 30.9% of adolescent girls use old closes for menstrual absorbent [23], 62.5% washed their genitalia using water only [24], 65.3% disposed absorbent material in open filled [12], and the proportion of adolescent girls practicing safe menstrual hygiene management inconstant throughout the country: 35.4% in Northeast Ethiopia [7], 29.8% in Northwest Ethiopia [25], 55.8% in Eastern Ethiopia [26], and 70.2% in Central Ethiopia [12]. Furthermore, adolescent girls in Ethiopia face additional burdens and challenges as a result of poor school water, sanitation, and hygiene (WASH) conditions (barriers include a lack of access to sanitation products and facilities, schools' lack of access to consistent supplies of water, and restrooms that are not MHM friendly), gender-based violence, and the stigmatization of menstruation [3, 27]. As a result, schoolgirls reported frequent episodes of low school attendance and absenteeism ranging from 19.2 to 54.5 percent [7, 27].

Several studies have been conducted across Ethiopia, examining the status of MHM among adolescent schoolgirls [7, 12, 13, 19, 23–26]. However, there was no nationwide study assessing the pooled proportion of safe MHM among adolescent girls. Further, the findings of these studies were inconsistent and characterized by significant variability. Therefore, this systematic review and meta-analysis aim to estimate: (1), the pooled proportion of safe MHM, and (2) the pooled estimate for the type of absorbent used by adolescent girls, disposal practice of absorbents, hygiene during menstruation, bathing during menstruation, and school absenteeism were measured using available studies. The findings of this study would be useful to policy-makers and program planners in the design of appropriate interventions to improve safe MHM in the country and similar settings.

## Methods

### Study design

A systematic review and meta-analysis were conducted to estimate the pooled proportion of safe MHM practice and its associated factors among adolescent girls in Ethiopia. This review and meta-analysis were conducted according to the guideline of Preferred Reporting Items for Systematic reviews and Meta-Analysis (PRISMA) [28] (**S1 File**). The protocol for this review was registered in the International Prospective Register of Systematic Reviews (PROSPERO), the University of York Centre for Reviews and Dissemination (record ID: CRD42021261351, 18th July, 2021)

### Searching strategy

We searched the following databases: PubMed/MEDLINE, Google Scholar, African Journal Online (AJOL), Hinari, Science Direct, ProQuest, Directory of Open Access Journals, POPLINE, and Cochrane Library inception to May 31, 2021. We used keywords: "Adolescent [MeSH Terms], OR Adolescents [Text Word], OR adolescence, OR puberty, OR peer, OR school" AND "Menstruation [MesH], OR menstrual, OR menses" AND "Hygiene [MesH], OR hygiene, OR hygienically, OR sanitation, OR sanitary, OR Feminine Hygiene Products [MesH], OR Menstrual Hygiene Products [MesH]" AND "Ethiopia" separately or in combination with the Boolean operator's terms "AND" and "OR" (**S2 File**). The electronic database search was also supplemented by searching for grey literature through Google scholar, Google

searching, and Ethiopian University digital repositories (such as the Addis Ababa University Digital Library). We also scan the reference lists of included studies for relevant studies.

## Study selection criteria

**Inclusion criteria.** *Study designs*. Both cross-sectional and cohort studies with baseline measures for the outcome of interest (menstrual hygiene practice/management) were eligible for this systematic review and meta-analysis.

*Study setting*. Only studies conducted in Ethiopia.

*Population*. Adolescent girls (age 10–19).

*Publication status*. Both published and unpublished studies were considered.

*Study period*. There was no restriction on the publication date.

*Language*. Articles published in the English language were considered

*Year of publication*. All publications reported up to May 31, 2021, were considered.

**Exclusion criteria.** Systematic reviews, commentaries, letters to editors, short communications, and qualitative studies were excluded. Also, articles that were not fully accessible after two-email contact with the primary/corresponding author were excluded. Studies that were not conducted in the adolescent age group were excluded.

## Study selection and data extraction

Two investigators (BS and DA) independently screened identified articles by their title, abstract, and full text to identify eligible articles against predetermined inclusion and exclusion criteria. The screened articles were compiled together from the two investigators (BS and DA) and discrepancies were resolved by consensus. We extracted data on study characteristics (such as study area, year of study, participants, age, sample size, response rate, and proportion of MHM). Endnote reference manager software was utilized to collect and organize search outcomes and for the removal of duplicate articles.

## Outcome measurement

The primary outcome of this study was safe menstrual hygiene practice. Adolescent girls using a clean menstrual management material to absorb or collect blood that can be changed in privacy as often as necessary for the duration of the menstruation period. The use of soap and water for washing the body as required and having access to facilities to dispose of used menstrual management materials termed as safe practice. We included studies that fulfilled the above definition. However, we found studies that used related definitions for safe menstrual hygiene practice. For this reason, we carefully checked the operational definition used by each included study and describe it in a table along with its proportion. In addition, the following outcomes were also examined: type of menstrual absorbent used by adolescent girls, disposal practice of used menstrual absorbents, daily bathing during menstruation, school absenteeism during menstruation, and change frequency of menstrual absorbent material.

## Operational definition

**Safe menstrual hygiene management practice.** Adolescent girls using a clean menstrual management material to absorb or collect blood that can be changed in privacy as often as necessary for the duration of the menstruation period, using soap and water for washing the body as required and having access to facilities to dispose of used menstrual management materials.

**Menstrual absorbents.** Refer to any product used by adolescent girls to collect menstrual flow; commercial pads refer to those marketed and available for purchase in shops.

## Quality assessment of the studies

The qualities of the included studies were assessed, and the risks for biases were refereed using the Joanna Briggs Institute (JBI) quality assessment tool for the prevalence studies [29]. Two reviewers (BS and DA) assess the quality of included studies independently. The assessment tool consists of nine parameters: (1) appropriate sampling frame, (2) proper sampling technique, (3) adequate sample size, (4) study subject and setting description, (5) sufficient data analysis, (6) use of valid methods for the identified conditions, (7) valid measurement for all participants, (8) using appropriate statistical analysis, and (9) adequate response rate. Failure to satisfy each parameter was scored as 1 if not 0. When the information provided was not adequate to assist in making a judgment for a specific item, we agreed to grade that item as 1 (failure to satisfy a specific item). The risks for biases were classified as either low (total score, 0 to 2), moderate (total score, 3 or 4), or high (total score, 5 to 9). Finally, articles with low and moderate quality were considered in this review (**S3 File**).

## Statistical methods and analysis

The extracted data were imported into STATA version 14 software for statistical analysis. The heterogeneity among all included studies was assessed by $I^2$ statistics and the Cochran Q test. The tests indicate significant heterogeneity among included studies ($I^2$ = 91.6, P-value < 0.001). Thus, a random-effects meta-analysis model was computed using the DerSimonian and Laird Method [30]. The overall pooled estimate was computed using the metaprop STATA command. The standard errors were calculated from the reported estimates and population denominators using a binominal distribution assumption. Accordingly, the pooled proportion of MHM and their corresponding 95% CI were presented using forest plots and tables. We performed a subgroup analysis based on geographical regions of the country, sample size, sampling methods, and publication year. Further, statistical analyses such as meta-regression and sensitivity analysis were performed to identify the possible sources of heterogeneity.

## Publication bias

In this meta-analysis, the presence of publication bias was evaluated using funnel plots and Begg's tests at a significance level of less than 0.05.

## Sensitivity analysis

Sensitivity analysis using a random-effects model was performed to assess the influence of a single study on the overall pooled estimate.

# Results

## Description of included studies

Overall, the searches identified 1045 articles (PubMed (n = 600), Google Scholar (n = 58), African Journal Online (n = 68), Hinari (n = 16), Science Direct (n = 39), ProQuest (n = 155), Direct of Open Access Journals (n = 11), POPLINE (n = 75), Cochrane Library (n = 12), and (n = 11) studies were identified through manual search) and 777 duplicates were removed. We screened the titles and abstracts of 268 articles and obtained 39 full texts, of which 22 studies [7, 12, 13, 19–21, 23–27, 31–41] met the inclusion criteria and included in the final analysis (**Fig 1**).

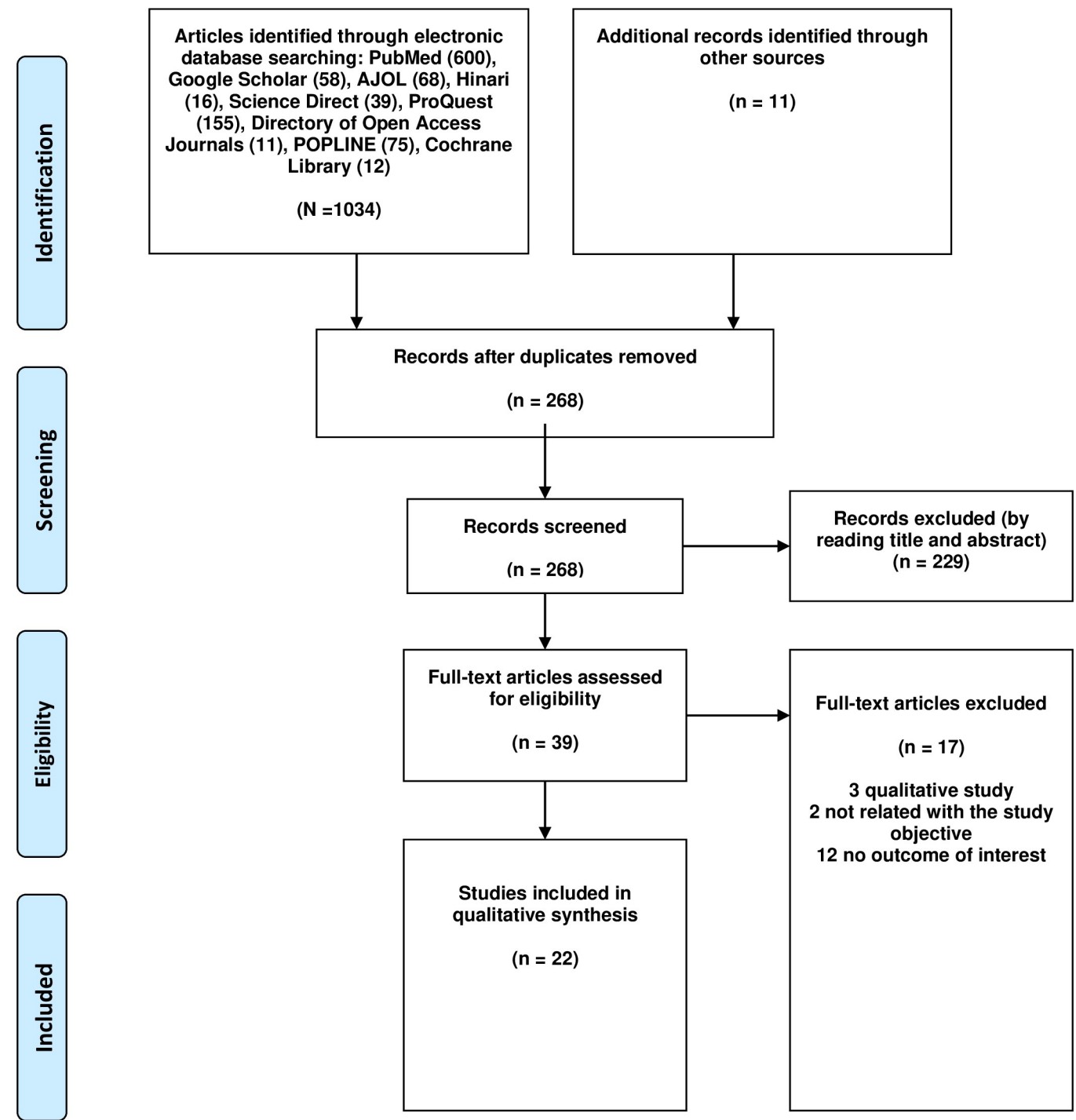

**Fig 1. Flow chart of study selection for systematic review and meta-analysis of menstrual hygiene management and associated factors among adolescent girls in Ethiopia.**

## Characteristics of included studies

As described in **Table 1**, 22 studies met the inclusion criteria and were included. All the included studies were cross-sectional studies by design. A total of 12,330 adolescent girls

**Table 1.  Descriptive summary of twenty-two studies included in the meta-analysis of MHM and associated factors among adolescent girls in Ethiopia (2004–2021).**

| Author's name, year of publication | Study area | Region | Study design | Methods of data collection | Sampling | Sample size | Response Rate | Safe MHM | Mean (±SD) age study participants | Mean age of menarche (±SD) | Quality |
|---|---|---|---|---|---|---|---|---|---|---|---|
| Birhane AD et al., 2020 [35] | Sebeta | Oromia | CS | Interviewer administered questionnaire | Simple random sampling | 466 | 95.9 | 49.1 | 15.5 ± 1.1 | 13.3 ± 1.0 | Low |
| Felleke AA et al., 2021 [26] | Harari | Harari | CS | Self-administered structured questionnaire | Systematic sampling | 301 | 100 | 55.8 | 15.87 | | Moderate |
| Kitesa B et al., 2016 [12] | Shoa zone | Oromia | CS | Interviewer administered questionnaire | Multi-stage sampling | 430 | 100 | 70.2 | | 13.69 ±1.18 | Low |
| Zeleke B, 2016 [41] | Bahir Dar | Amhara | CS | Self-administered structured questionnaire | Simple random sampling | 649 | 95 | 84.3 | 17.29 ±1.43 | | Low |
| Biruk E, 2017 [24] | Addis Ababa | Addis Ababa | CS | Self-administered structured questionnaire | Multi-stage sampling | 756 | 98 | 52.5 | 14.89 ± 1.28 | 12.84 +0.74 | Low |
| Bekele F et al., 2017 [34] | Batu | Oromia | CS | Interviewer administered questionnaire | Systematic sampling | 274 | 100 | 66.8 | 15.72 ± 1.32 | | Moderate |
| Gedefaw G et al., 2019 [36] | Woldia | Amhara | CS | Not clearly described | Systematic sampling | 409 | 96.7 | 48.9 | 16.4 ± 1.5 | | Moderate |
| Bulto GA, 2019 [19] | Holeta | Oromia | CS | Self-administered structured questionnaire | Systematic sampling | 403 | 97.1 | 37.4 | | 13.62 ± 1.47 | Low |
| Anchebi HT et al., 2017 [32] | Adama | Oromia | CS | Self-administered structured questionnaire | Multistage stratified sampling | 398 | 94.3 | 57 | | | Low |
| Kedir T, 2017 [23] | West Shoa | Oromia | CS | Interviewer administered questionnaire | Multistage stratified sampling | 610 | 98.7 | 45.6 | 16.07 ± 1.65 | | Low |
| Fisseha MA et al., 2017 [25] | Wegera | Amhara | CS | Self-administered structured questionnaire | Simple random sampling | 423 | 100 | 29.8 | 17 ±1.5 | 15 ± 1.08 | Low |
| Azage M et al., 2018 [33] | Bahir Dar | Amhara | CS | Interviewer administered questionnaire | Multi-stage stratified sampling | 1006 | 99.6 | 24.5 | 17.27±1.43 | | Low |
| Gena HM, 2017 [37] | East Harerge | Oromia | CS | Interviewer administered questionnaire | Simple random sampling | 672 | 99.4 | 58.3 | 16.5 ±1.12 | | Low |
| Abebe M, 2017 [27] | Akaki | Addis Ababa | CS | Self-administered structured questionnaire | Stratified sampling technique | 1065 | 100 | 45.2 | 16.49 ±1.23 | | Low |
| Niguse R et al., 2019 [39] | Goba | Oromia | CS | Interviewer administered questionnaire | Systematic sampling | 282 | 98.6 | 71.4 | 16.48 ±0.86 | | Moderate |
| Shallo SA et al., 2018 [13] | Ambo | Oromia | CS | Self-administered structured questionnaire | Simple random sampling | 336 | 91.5 | 46.7 | 16± 2.25 | 13.9 ± 0.71 | Low |
| Upashe SP et al., 2015 [20] | Nekemte | Oromia | CS | Self-administered structured questionnaire | Simple random sampling | 828 | 98 | 39.9 | | | Low |
| Gultie TK, 2014 [38] | Mahelmeda | Amhara | CS | Self-administered structured questionnaire | Multi-stage sampling | 492 | 100 | 90.8 | 16.85± 1.336 | | Low |

(*Continued*)

**Table 1.** (Continued)

| Author's name, year of publication | Study area | Region | Study design | Methods of data collection | Sampling | Sample size | Response Rate | Safe MHM | Mean (±SD) age study participants | Mean age of menarche (±SD) | Quality |
|---|---|---|---|---|---|---|---|---|---|---|---|
| Abera Y, 2004 [31] | Addis Ababa | Addis Ababa | CS | Self-administered structured questionnaire | Multi-stage sampling | 829 | 96.1 | 47.6 | 16.13± 1.57 | 16.13± 1.57 | Low |
| Tegegne TK, 2014 [7] | Habru | Amhara | CS | Self-administered structured questionnaire | Multi-stage sampling | 455 | 96.4 | 35.4 | 14.96 (±1.33) | | Moderate |
| Belayneh Z et al., 2019 [21] | Gedeo | SNNPR | CS | Interviewer administered questionnaire | Multi-stage sampling | 791 | 98.1 | 39.7 | 16.3 ± 4.7 | | Low |
| Shumie ZS, 2021 [40] | South Wollo | Amhara | CS | Self-administered structured questionnaire | Simple random sampling | 441 | 100 | 62.4 | 17.72 ± 1.49 | 13.92 ± 0.92 | Low |

CS: Cross-sectional; SNNPR: South Nations and Nationalities People of the Region; MHM: Menstrual hygiene management; SD: Standard Deviation.

participated in the current systematic review and meta-analysis. The median response rate of the included studies was 98.35%. The included twenty-two studies were published from 2004 to 2021. In the current meta-analysis, the sample size varied from small (n = 274) [34] to large (n = 1,065) [27]. From the reports of primary studies, the mean age of the respondents ranged from 14.9 ±1.3 [24] to 17.7±1.5 [40] years. In this review, the lowest proportion (24.5%) of safe MHM was found in a study conducted at the Bahir Dar, Amhara region [33], while the highest proportion (90.8%) of safe MHM was reported in a study conducted at Mahelmeda, Amhara region [38]. These 22 studies were conducted in four Ethiopian regions and one administrative town (Addis Ababa). Ten studies were conducted in Oromia [12, 13, 19, 20, 23, 32, 34, 35, 37, 39], seven in Amhara region [7, 25, 33, 36, 38, 40, 41], three in Addis Ababa [24, 27, 31], one in SNNPR (South Nations and Nationalities People of the Region) [21] and one Harari regions [26]. No studies were reported from Tigray, Benishangul Gumiz, Afar, Somali, Gagmbela regions, and Dire Dawa city administrations. Concerning the quality score of included studies, most articles (77.3%) had low-quality scores (**Table 1** **and S3 File**). In addition, 18 studies [7, 12, 13, 19–21, 23–26, 32, 33, 35–38, 40, 41] performed multivariate analyses to identify factors associated with MHM in Ethiopia (**Table 2**).

## Meta-analysis of pooled safe MHM in Ethiopia

The overall pooled proportion of safe MHM in Ethiopia was found to be 52.69% (95%CI: 44.16, 61.22) (**Fig 2**). The included studies exhibited significant heterogeneity ($I^2$ = 99.1%, p < 0.001), which led us to compute a random effect meta-analysis model to estimate the pooled proportion of safe MHM in Ethiopia.

In this meta-analysis, we performed subgroup analysis based on the studies' geographical setting (Region of the country). Accordingly, the highest proportion of safe MHM was observed in Harari region with proportion of 55.8% (95% CI: 50.19, 61.41%), followed by Oromia region with proportion of 54.19% (95%CI: 46.61, 61.77%), Amhara region 53.75% (95% CI: 30.83, 76.66%), Addis Ababa 48.35% (95%CI: 44.17, 52.53%), and the least proportion was reported from SNNP region 39.70% (95%CI: 36.29, 43.11%) (**Fig 3**). Concerning sample size, the proportion of safe MHM was higher in studies having a sample size ≤ 400 adolescent girls, 59.53% (95% CI: 51.04, 68.02%) compared to those having a sample size > 400 adolescent girls, 50.69% (95% CI: 40.47, 60.92%) (**Table 3**).

**Table 2. Summarized description of studies on factors associated with menstrual hygiene management.**

| Author | Region | Study design | Operation definition used for outcome variable (safe menstrual hygiene) | Safe MHM (%) | Identified factors associated with MHM |
|---|---|---|---|---|---|
| Birhane AD et al., 2020 [35] | Oromia | A school-based cross-sectional | In the current study, when an adolescent girl during menstruation changes menstrual pad and washes perinea area at least three times per day, it is considered as „ideal" practice, if the practice is done twice per day it is considered as „good" practice otherwise it is considered as „fair" practice. However, if an adolescent girl does not change sanitary pads and wash the perineal area within a day, she is considered as not practicing menstrual hygiene. | 45.6 | Students whose mothers can read write were more than 4 fold likely to practice menstrual hygiene (AOR = 4.34 with a 95% CI [1.91, 9.86]) as compared to girls whose mothers are illiterate. Adolescent girls with unfavorable attitudes towards menstrual hygiene management were 63% less likely to practice menstrual hygiene management (AOR = 0.37 with a 95% CI [0.22, 0.61]) as opposed to adolescent girls with a favorable attitude. Adolescent in-school girls with partial awareness about menstruation and its management were 40% less likely to practice the hygiene practice (AOR = 0.60 with a 95% CI [0.36, 0.98]) when compared to those with a good level of awareness about menstruation and its management. |
| Felleke AA et al., 2021 [26] | Oromia | A school-based cross-sectional study | A composite score was used, and the detailed description of the outcome definition was not clearly described. | 55.8 | The students who live in the rural area (AOR = 0.27; 95% CI: 0.12, 0.58) were 73% less likely to had good menstrual hygiene practice than students who live in the urban, students who have no permanent pocket money from family (AOR = 0.36, 95% CI: 0.31, 0.99) were 64% less likely have good menstrual hygiene practice than students who have permanent pocket money from family. Students live with uneducated fathers (AOR = 0.39, 95% CI: 0.18,0.87) were 61% less likely to have good menstrual hygiene practice than those with educated father. |
| Kitesa B et al., 2016 [12] | Oromia | A school-based cross-sectional study | A composite score was used and items used to assess hygiene practice includes kinds of sanitary pad used, frequency of changing sanitary na[kins, means of managing emergency bleeding while in school, disposal practice of sanitary napkins, how to clean the body while on menstruation, and frequency of bathing genitalia area. | 70.2 | Students who were in grade 11 were 2.84 times more likely to practice good menstrual hygiene than their counterparts who were in grade nine [AOR = 2.84, 95%CI: 1.35–5.97]. Those accessed to WASH facilities (to water supply and clean and privacy secured toilet) were 3.4 times more likely to practice good menstrual hygiene management times than their counters [AOR = 3.4, 95% CI = (2.35–4.79)] |
| Gena HM, 2017 [37] | Oromia | A school-based cross-sectional study | A composite score was used and items used to assess hygiene practice includes: Use of absorbent materials during menstruation; Use of commercial disposable sanitary pads; Changes pads or clothes more than three times a day during menstruation; Cleans reusable clothes with soap and water; Dry reusable clothes in the sunlight; Disposes pads by wrapping with paper; Takes bath daily with soap during menstruation; Cleans external genitalia during menstruation, and Cleans external genitalia with water and soap during menstruation. Disposes of used sanitary pads in dustbins | 58.3 | Girls from urban areas were 2.59 times more likely to have good menstrual hygiene management practices compared to their counterparts (AOR = 2.59, 95% CI: 1.77–3.80). Girls whose mother's educational status was secondary school and above were about 2 times more likely to have good menstrual hygiene management practices compared to those whose mothers had no formal education (AOR = 1.83, 95% CI: 1.01–3.30). The likelihood of practicing good menstrual hygiene management was 2.78 times higher among those students who had moderate knowledge (AOR = 2.78, 95% CI: 1.64–5.28) and 3.87 times more likely among those who had good knowledge (AOR = 3.87, 95% CI: 2.21–6.77) compared to those who had poor knowledge |
| Biruk E, 2017 [24] | Oromia | A school-based cross-sectional study | Adolescent school girls using a clean menstrual management material to absorb or collect blood that can be changed in privacy as often as necessary for the duration of the menstruation period, using soap and water for washing the body as required, and having access to facilities to dispose of used menstrual management materials | | Students above fifteen years old were 2.28 [AOR: 2.28, 95% CI: 1.61–4.97] times more likely to have good menstrual hygiene practice than their counterparts. Girls whose mother's education secondary and above were eight times more likely to have good practice about menstrual hygiene compared to those from illiterate mothers [AOR = 7.76, 95% CI: 3.58–16.81]. Girls whose fathers from private employees were [AOR = 3.65, 95% CI: 1.21–10.99] times more likely to have good menstrual hygiene management practice than those who were daily laborer families. This study also found that girls whose age at first menarche greater than thirteen were 2.57 times more likely to have good practice about menstrual hygiene compared to those who were less than thirteen years old [AOR = 2.57, 95% CI: 1.41–4.69]. Knowledge of the respondents towards menstrual hygiene management was significantly associated with their practice [AOR = 4.58, 95% CI: 2.46–8.53]. |
| Fisseha MA et al., 2017 [25] | Amhara | A school-based cross-sectional study | The measurement of the practice of menstrual hygiene focuses on the use of material during menstruation, methods of disposal of materials (use of sanitary pad), cleaning of external genitalia (cleaning 2 or more times/day), frequency of sanitary pad change (changing pad 2 or more times/day) and materials used for cleaning purpose (washing with soap and water or with plain water). | 29.8 | The likelihood of good menstrual practice among girls who had exposure to advertisement is two times higher compared to girls who had no exposure to the advertisement (AOR = 2.06, 95%CI: 1.27, 3.34). Girls with good knowledge of menstrual hygiene were two times more likely to have good practice when compared to girls with poor knowledge (AOR = 2.23, 95%CI: 1.06, 4.71). |

(Continued)

Table 2. (Continued)

| Author | Region | Study design | Operation definition used for outcome variable (safe menstrual hygiene) | Safe MHM (%) | Identified factors associated with MHM |
|--------|--------|--------------|--------------------------------------------------------------------------|--------------|----------------------------------------|
| Gultie TK, 2014 [38] | Amhara | A school-based cross-sectional study | The measurement of the practice of menstrual hygiene focuses on the use of material during menstruation, methods of disposal of materials (use of sanitary pad), cleaning of external genitalia (cleaning 2 or more times/day), frequency of sanitary pad change (changing pad 2 or more times/day) and materials used for cleaning purpose (washing with soap and water or with plain water). | 90.9 | It revealed that the practice of good menstrual hygiene was more among students who live in the urban (AOR 2.38: 95% CI, 1.14, 3.05) than those students who live in the rural area, whose source of information was a teacher (AOR 7.64: 95% CI, 2.16, 27.03) than students whose source of information was mother, who have access for water (AOR 6.504: 95% CI, 2.08, 20.32) were more practiced good menstrual hygiene than those who didn't have access for water, who had a high level of knowledge about menstrual hygiene (AOR 5.78: 95% CI, 2.16, 15.51) than those students who had a low level of menstrual hygiene knowledge |
| Azage M et al., 2018 [33] | Amhara | Community-based cross-sectional study | Adolescent girls used menstruation pads, wash their genitalia two or more times per day, and disposed of used menstruation pad into a latrine, their menstrual management hygiene practice | 24.5 | Adolescent girls whose age is >18years of age were 1.4 times more likely to have safe menstrual hygiene management practice than their counterparts [AOR = 1.46,95% CI: (1.1, 1.9)]. The educational status of adolescent girls and their mothers had associations with their menstrual hygienic practice. Adolescent girls who attended primary education were 5 times [AOR = 5.01, 95% CI: (2.5, 9.7)], those who attended secondary education were 8.5 times [AOR = 8.53, 95% CI: (4.4, 7.89, 16.4)] and those attend college and above were 6.9 times [AOR = 6.96, 95% CI: (3.1, 15.4)] more likely to have safe menstrual hygiene management practice than those who had no formal education. Adolescents whose mother was able to read and write were 3 times more likely to have safe menstrual hygiene management practice [AOR = 3.14, 95% CI: (1.7, 5.5)] than those mothers who could not able to read and write. Mothers of adolescent girls who attended primary education [AOR = 3.29, 95% CI: (1.9, 5.5)] and secondary and above [AOR = 3.62, 95% CI: (2.1, 6.0)] were 3.2 times and 3.6 times, respectively, more likely to have safe menstrual hygiene management practice than those mothers who could not able to read and write. Being a student currently and adolescents who believe menses needs special care were factors associated with safe menstrual hygiene practice. Adolescents who are a student currently were 1.8 times and those who believe menses needs special care were 3.2 times more likely to have safe menstrual hygiene management practice than their counterparts [AOR = 1.80, 95% CI: (1.2, 2.5)] and [AOR = 3.21, 95% CI: (1.1, 10.9)] respectively. |
| Tegegne TK, 2014 [7] | Amhara | School-based cross-sectional study | Use of menstrual absorbent during their last menstruation. | 35.4 | Students who were living in urban areas were 2.32 times more likely to use a disposable sanitary napkin than their counterparts [AOR (95% C.I) 2.32 (1.21–4.45)]. Girls from literate families; i.e. who can read and write, completed primary and secondary education [AOR (95% C.I): 2.30 (1.23–4.30), 4.13 (1.85–9.18), 4.26 (1.61–11.28)] respectively were more likely to use sanitary napkins than their counterparts who were from illiterate family. Girls from families having a household monthly expenditure of 601–900, 901–1200 and greater than 1200 Ethiopian Birr [AOR (95% C.I): 3.24 (1.46–7.17), 3.41 (1.56–7.43), 4.97 (2.21–11.16)] respectively were more likely to use sanitary napkins than their counterparts expending less than 600 Ethiopian Birr. Schoolgirls who were living with relatives [AOR (95% C.I): 0.16 (0.04–0.56) were less likely to use disposable sanitary napkins than their counterparts |

(Continued)

**Table 2.** (Continued)

| Author | Region | Study design | Operation definition used for outcome variable (safe menstrual hygiene) | Safe MHM (%) | Identified factors associated with MHM |
|---|---|---|---|---|---|
| Bulto GA, 2019 [19] | Oromia | School-based cross-sectional study | Adolescent girls using a clean menstrual management material to absorb or collect blood that can be changed in privacy as often as necessary for the duration of the menstruation period, using soap and water for washing the body as required, and having access to facilities to dispose of used menstrual management materials. | 37.4 | The current study identified that students who were from the urban residence were 2.62 times more likely to have safe Menstrual Hygiene Management (MHM) practice than those from rural residences (AOR = 2.62, 95% CI: 1.53–4.48). Those adolescents' girls who got information about menstruation before menarche from their mothers (AOR = 2.17, 95% CI: 1.18–3.96) and at schools from teachers (AOR = 5.09, 95% CI: 2.67–9.67) were twice and five more likely to practice adequate/safe MHM than those who did not. Students whose school toilets had female toilets with inside locks were 2.82 times more likely to have safe or adequate MHM practices than those who did not have (AOR = 2.82, 95% CI: 1.67–4.76). Adolescent girls who missed their school for one day (AOR = 3.69, 95% CI: 1.55–8.77) and did not miss school (AOR = 4.2, 95% CI: 1.55–11.41) during their menses were 3.69 and 4.2 times more likely to practice safe MHM than those who missed more than one day respectively. Adolescent girls who experienced health problems during menstruation were 2.6 times more likely to practice adequate MHM than those who did not experience (AOR = 2.63, 95% CI: 1.49–4.64). Those students who ever experienced any whitish or gray discharge through the vagina were 2.8 times more likely to have adequate MHM practices than those who did not experience it (AOR = 2.84, 95% CI: 1.66–4.85). Those adolescent girls who had good overall knowledge about menstruation were almost twice more likely to practice adequate or safe MHM than those who had poor knowledge (AOR = 1.94, 95% CI: 1.07–3.52) |
| Anchebi HT et al., 2017 [32] | Oromia | School-based cross-sectional study | A composite score was used and items used to assess hygiene practice includes Use of absorbent materials during menstruation; type of material used; place of disposal; use of clean cloths, soap, and water; clean external genitalia; use of soap and water for bathing; and frequency of changing absorbents. | 57 | Students who don't felt the school was comfortable 44% less likely to had good menstrual hygiene practice than students who felt school was comfortable (AOR = 0.56; 95%CI = 0.37–0.85). Furthermore, students whose source of money was their parents were 2.3 times more likely to had a good menstrual hygiene practice as compared to those students who earn the money by themselves (AOR = 2.27; 95% CI = 1.07, 4.77). Regarding mother's educational status, students whose mother attended secondary and above level of education were 39% less likely to had good menstrual hygiene practice than students whose mother attended below and primary level of education (AOR = 0.61; 95%CI = 0.37–0.99). |
| Upashe SP et al., 2015 [20] | Oromia | School-based cross-sectional study | A composite score was used and items used to assess hygiene practice includes: Uses absorbent materials during menstruation; use of clean clothes with soap and water; Changing pads or clothes more than three times and above during menstruation; Takes bath daily with soap during menstruation; Cleans external genitalia with water and soap during menstruation, and Disposes used sanitary pads in a dustbin | 39.9 | Girls whose mother's educational status was secondary school and above were 2 times more likely to have good practice of menstrual hygiene than their counterparts [AOR = 2.03, 95% CI: 1.38–2.97]. Respondents whose mother's occupations come under the category of others (such as merchants and governmental and private employees) were less likely to have good practice of menstrual hygiene than housewives [AOR = 0.66, 95% CI: 0.47–0.91]. Girls who earn permanent pocket money from their families were nearly three times more likely to have good practice about menstrual hygiene compared to those who don't earn permanent pocket money from their families [AOR = 2.73, 95% CI: 1.76–4.26] |
| Shumie ZS, 2021 [40] | Amhara | School-based cross-sectional study | Menstrual hygiene management indicates the practice of secondary school girls using clean materials to absorb or collect menstrual blood that can be changed privately, safely, hygienically, and as often as necessary for the duration of the menstruation period, using soap and water for washing the body as required, and having access to facilities to dispose of used menstrual management materials. | 62.4 | Students who had good knowledge about menstruation and its hygienic management were 1.73 times more likely to practice good menstrual hygiene than their counterparts [AOR = 1.73, 95% CI (1.07–2.80)], who had poor knowledge about menstruation and its hygienic management. On the other hand, students who were living in urban areas were 3.76 times more likely to practice good menstrual hygiene than their counterparts who were living in rural [AOR = 3.76, 95% CI (2.18–6.51)]. Girls who know about RTIs / STIs were 2.46 times more likely to practice good menstrual hygiene than their counterparts who don't know about RTIs /STIs [AOR = 2.46, 95% CI (1.37–4.43)]. In addition to this, students whose parents having private showers were 2.04 times more likely to practice menstrual hygiene than their counterparts whose parents are not having private showers [AOR = 2.04, 95% C.I (1.24–3.37)]. |

(Continued)

**Table 2.** (Continued)

| Author | Region | Study design | Operation definition used for outcome variable (safe menstrual hygiene) | Safe MHM (%) | Identified factors associated with MHM |
|---|---|---|---|---|---|
| Zeleke B, 2016 [41] | Amhara | School based cross-sectional study | Menstrual hygiene focuses on the use of sanitary pad material during menstruation, washing with soap and water or with plain water; taking shower (>2 times /day). | 84.3 | Good menstrual hygiene was more among students who live in the urban (AOR 2.708:95% CI, 1.712, 4.285) than students who live in the rural area, students who have access to water (AOR 1.553:95% CI, 0.309, 0.989) than students who have no access for water, students from private schools (AOR 4.425:95% CI, 1.793, 10.924) than who is in the public schools and students who had heard about sanitary materials(AOR 2.493:95% CI, 1.478, 4.207) than who had not heard. |
| Kedir T, 2017 [23] | Oromia | School-based cross-sectional study | Adolescent girls using a clean menstrual hygiene material to absorb or collect blood that can be changed in privacy as often as necessary for the duration of the menstruation period, using soap and water for washing the body as required, and having access to facilities to dispose of used menstrual hygiene materials | 45.6 | Adolescent girls age between 13–15, 47% less likely to be good practice menstrual hygiene [AOR: 0.53, 95%CI (0.31, 0.89)] than those adolescent girls age between16-19. Adolescent girls who live in a rural area less likely to be good practice menstrual hygiene [AOR: 0.49,95%CI(0.28, 0.85)] than those adolescent girls who live in urban areas. Adolescent girls whose not come to school during menstruation 44.4% were less likely to be good practice menstrual hygiene[AOR: 0.55, 95%CI(0.34, 0.88)] than those who came to school during menstruation. Adolescent girls who don't have a place for drying absorbent material have 70.4% less likely to be good practice menstrual hygiene [AOR:0.29,95%CI(0.18,0.49)] than adolescent girls who have a place for drying absorbent materials. Adolescent girls not provided pocket money 38% less likely to be good menstrual hygiene [AOR: 0.62, 95%CI (0.39, 0.97)] than adolescent girls who provided permanent pocket money. Adolescent girls who asked advice friend 15.39 more likely to be good practice of menstrual hygiene [AOR:15.39, 95%CI (1.746, 135.825)] than adolescent girls who asked advice father, sister, teacher. |
| Belayneh Z et al, 2019 [21] | Southern Ethiopia | School-based cross-sectional study | A composite score was used and items used to assess hygiene practice includes: Using absorbent materials during menstruation; Change pads or clothes more than three times a day during menstruation; Using clean clothes with soap and water; Cleans external genitalia during menstruation; Washes bath daily with soap during menstruation; Cleaning of external genitalia with water and soap during menstruation, and Disposes used sanitary pads in a dustbin. | 39.7 | Adolescent girls aged < 15 age (AOR: 1.71; 95%CI:(1.22, 2.39); adolescent with longer duration (> 5 day) of menses flow (AOR: 2.51; 95%CI (1.66,3.80)); and those with good knowledge towards menses (AOR: 1.48; 95%CI(1.04,2.1)) had statistically significant association with the good menstrual hygiene practice. |
| Gedefaw G et al, 2019 [36] | Woldia | School-based cross-sectional study | Menstrual hygiene indicates personal hygiene during menstrual flow includes bathing once a day, changing clothes regularly, and changing pads at least two-four times per day. | 48.9 | Girls who had good knowledge of menstrual hygiene have 3.74 more likely good practices on menstrual hygiene as compared to Girls who had poor knowledge on menstrual hygiene (AOR = 3.74,95% CI(1.18–7.7). Respondents whose mother's level of education is secondary and above had 1.86 more likely good practice as compared with those whose mother's level of education is elementary and no formal education(AOR = 1.86,95%CI(1.18–2.9). Being a grade 10 students had 2.3 more likely had a good practice on menstrual hygiene as compared to grade 9 students(AOR = 2.3, 95%CI(1.48–3.56). |
| Shallo SA et al, 2018 [13] | Ambo | School-based cross-sectional study | The menstrual hygiene practice is considered to be safe if it fulfills all of the following four criteria, otherwise considered as unsafe: If the females used safe absorbents (considered safe if they were commercially available sanitary pads (locally called modes/TAMPON) or new clothes). Changing absorbents three or more times per 24 hours, If girls wash their genitalia two or more times per day and disposed of used menstruation pad by burying or if burnt it after use. | 46.7 | Compared to age greater or equal to 18 years, those females in the age of less than 18 years were less likely to had unsafe MHM practice [AOR: 0.16, 95%CI (0.045, 0.57)]. Students whose fathers were attended higher education (first degree and above) were less likely [AOR: 0.28, 95% CI: (0.10, 0.88)] less likely their menses is unsafely managed compared to those whose father can't read or write. Compared to those females who reported that they used media (electronic/books) as a common source of information about menses related issues, those who claimed their school teachers were the common source of information on the menstrual issue were 3.75 times more likely to have unsafe MHM practice (AOR:3.75, 95% CI: 1.75). |

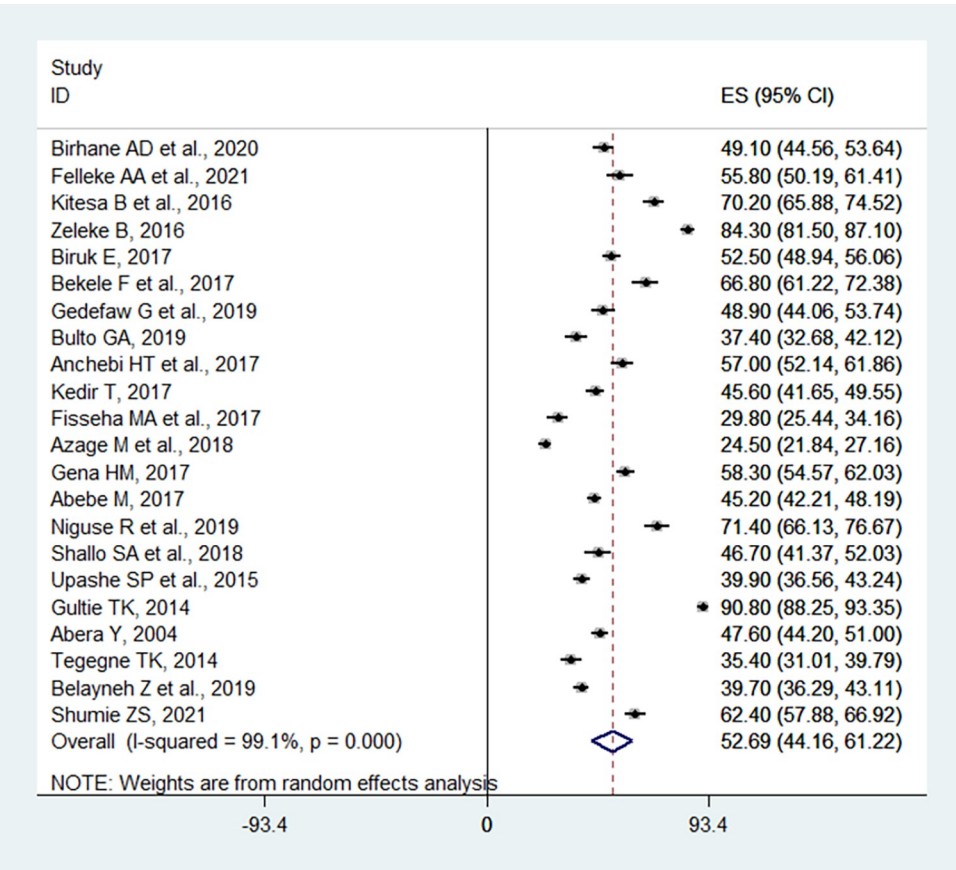

**Fig 2. Forest plot of the pooled proportion of safe MHM among adolescent girls in Ethiopia, studies published between 2004 and 2021.**

## Type of menstrual absorbents use

The use of commercial menstrual absorbents in Ethiopia was 64.63% (95%CI: 55.32, 73.93, $I^2$ 99.2%, n = 18). The use of commercial absorbents was different by regions, the highest commercial pads use was reported in Addis Ababa, 87.6% (95%CI: 85.62, 89.58,), followed by Harari region 72.8% (95%CI: 67.77,77.82), Oromia region 68.1% (95%CI: 56.68,79.52), Amhara 59.38% (95%CI: 41.59,77.18) and the least was reported from SNNP region 42.4% (95%CI: 38.95,45.84). Homemade cloth and disposable piece of rags use were common in studies conducted in Amhara region 53.03% (95%CI: 22.29,83.77) and 14.33% (95%CI: 7.20,21.46), respectively (**Table 4**).

## Hygiene during menstruation

In the current systematic review, the majority of adolescent girls wash their genitalia during menstruation 88.18% (95% CI: 83.82–92.53, 91%, $I^2$ 98.4%). However, 50.69% (95%CI: 41.53, 59.84, $I^2$ 98.9%) of schoolgirls reported they used both soap and water for genital cleaning, and 45.85% (95%CI: 39.47, 52.23, $I^2$ 95.9%) of schoolgirls only used water for genital cleaning. Regarding the frequency of genital cleaning, less than half [41.67% (95%CI: 27.12–56.23, $I^2$ 98.8%)] of schoolgirls reported wash their genitalia daily or two-time a day during menstruation (**Table 4**). A subgroup of hygiene during menstruation found that daily washing was less common in the Harari region 27.9% (95%CI: 22.83,32.97) and higher in the Oromia region

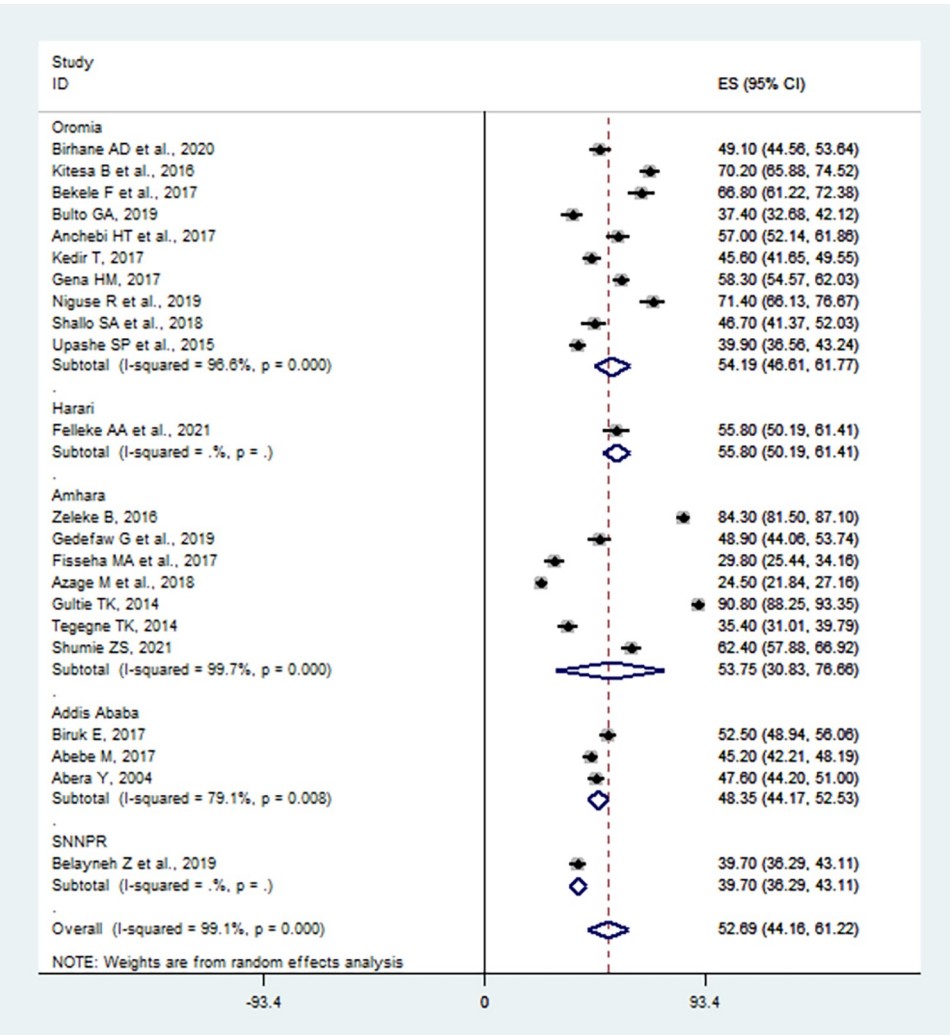

**Fig 3. Subgroup analysis of menstrual hygiene management among adolescent girls by Ethiopian regions, studies published between 2004 and 2021.**

51.84% (95%CI: 17.74,85.93). On the other hand, the use of both soap and water for genital washing was higher in the SNNP region 69.50% (95%CI: 66.29,72.71), and lower in Addis Ababa 29.5% (95%CI: 26.24,32.75) (**Table 4**).

## Disposal of absorbent material

Disposal of absorbent material into the latrine was the most common practice in Ethiopia (PP = 62.18%, 95%CI:52.87,71.49, $I^2$98.7%). This practice was comparable in all regions of Ethiopia (Oromia 65.38%, Amhara 62.44%, Harari 54.20%, and Addis Ababa 52.40%). About 26.96% (95%CI: 16.52, 37.41) and 27.35% (95%CI: 18.55,36.14) of schoolgirls put absorbent in dust bins and disposed of by wrapping with paper, respectively. Also, one-sixth, 15.85% (95% CI: 9.23,22.46, $I^2$99.3%) of girls threw the absorbent in open field; this was higher in Addis Ababa 46.20% (95%CI: 43.21,49.19). Burying (6.6%) and burning (11.8%) of absorbent material was the least reported practice by adolescent girls (**Table 4**).

**Table 3. Subgroup analysis of menstrual hygiene management among adolescent girls in Ethiopia, studies published between 2004 and 2021.**

| Sub-Group | Number of studies (n) | Proportion of safe MHM | 95%CI | $I^2$ (%) | p-value |
|---|---|---|---|---|---|
| **Regions** | | | | | |
| Oromia | 10 | 54.19 | 46.60–61.77 | 96.6 | p<0.001 |
| Amhara | 7 | 53.75 | 30.83–76.66 | 99.7 | p<0.001 |
| Addis Ababa | 3 | 48.35 | 44.17–52.53 | 79.1 | P = 0.008 |
| SNNPR | 1 | 39.70 | 36.29–43.11 | - | - |
| Harari | 1 | 55.80 | 50.19–61.41 | - | - |
| **Study period** | | | | | |
| ≤ 2015 | 4 | 53.45 | 25.15–61.23 | 99.7 | p<0.001 |
| >2015 | 18 | 52.51 | 44.18–60.85 | 98.7 | p<0.001 |
| **Sample size** | | | | | |
| ≤400 | 5 | 59.53 | 51.04–68.02 | 92.1 | p<0.001 |
| >400 | 17 | 50.69 | 40.47–60.92 | 99.3 | p<0.001 |
| **Methods of data collection** | | | | | |
| Interviewer Administered | 8 | 53.13 | 40.52–65.72 | 98.8 | p<0.001 |
| Self-administered | 13 | 52.70 | 44.00–61.75 | 99.2 | p<0.001 |
| **Sampling methods used** | | | | | |
| Simple random sampling | 7 | 52.95 | 37.58–68.33 | 99.1 | p<0.001 |
| Systematic sampling | 5 | 56.02 | 43.66–68.37 | 96.5 | p<0.001 |
| Multi-stage sampling | 6 | 56.05 | 36.93–75.17 | 99.4 | p<0.001 |
| Multi-stage stratified sampling | 3 | 42.29 | 22.62–61.96 | 98.8 | p<0.001 |
| Stratified sampling | 1 | 45.20 | 42.21–48.19 | - | - |
| **Risk of bias** | | | | | |
| Low | 17 | 51.83 | 41.67–61.99 | 99.3 | p<0.001 |
| Moderate | 5 | 55.60 | 42.34–68.87 | 97.0 | p<0.001 |

## Bathing during menstruation

Almost half (52.59%, 95% CI: 40.45, 64.73%, $I^2$ 99.2%) bathed daily during menstruation. A subgroup of studies found that daily baths were more common in Addis Ababa 66.6% of studies (95%CI: 63.03,70.17) (**Table 4**).

## School absenteeism during menstruation

In the current study, seven studies (7,12,13,23,27,31,33) reported on school absenteeism associated with menstruation, with one in four girls missing one or more school days during menstruation (PP: 32.03%, 95%CI: 22.65%, 41.40%, $I^2$ 98.2%), this was higher in Amhara region 37.89% (95%CI: 5.45,70.33), followed by Addis Ababa 31.72% (7.12,56.32) and Oromia region 28.38% (19.35,37.41) (**Table 4**).

## Sensitivity analyses

To detect the influence of one study on the overall meta-analysis estimate, sensitivity analysis using a random-effects model was used, and there is no evidence for the influence of a single study on the overall pooled result of safe MHM in Ethiopia (**Fig 4** and **S4 File**).

## Meta-regression

We conducted a univariate meta-regression model by considering publication year, sample size, and quality score as covariates to identify the possible sources of heterogeneity across

**Table 4. Pooled proportion of absorbents used, hygiene during menstruation, the disposal of the absorbent, use of a daily bath during menstruation, and school absenteeism during menstruation in Ethiopia studies published between 2004 and 2021.**

| Factors | Total Pooled | | | | Sub-group by region | | | | | | | | | | |
|---|---|---|---|---|---|---|---|---|---|---|---|---|---|---|---|
| | | | | | Oromia | | Amhara | | Harari | | Addis Ababa | | SNNPR | | |
| | # | PP, 95% CI | I², % | p-value | # | Proportion, 95% CI | # | Proportion, 95% CI | # | Proportion, 95% CI | # | Proportion, 95% CI | # | Proportion, 95% CI | |
| *Absorbents used* | | | | | | | | | | | | | | | |
| Commercial pads | 18 | 64.63 [55.32, 73.93] | 99.2 | p<0.001 | 8 | 68.10 [56.68,79.52] | 7 | 59.38 [41.59,77.18] | 1 | 72.80 [67.77,77.82] | 1 | 87.60[85.62, 89.58] | 1 | 42.40 [38.95,45.84] | |
| Disposable piece of rags | 12 | 10.07 [6.63,13.50] | 96.1 | p<0.001 | 6 | 7.48 [3.79,11.17] | 5 | 14.33 [7.20,21.46] | 1 | 4.70 [2.31,7.09] | - | - | - | - | |
| Use paper/toilet paper/ underwear | 9 | 5.83 [3.66,8.01] | 91.4 | p<0.001 | 3 | 7.28 [1.78,12.77] | 5 | 4.45[2.14,6.76] | 1 | 9.30 [6.02,12.58] | - | - | - | - | |
| Homemade cloth | 3 | 53.03 [22.29,83.77] | 99.4 | p<0.001 | - | - | 3 | 53.03 [22.29,83.77] | - | - | - | - | - | - | |
| *Hygiene during menstruation* | | | | | | | | | | | | | | | |
| Washing their genitalia during their menstruation | 12 | 88.18 [83.82,92.53] | 98.4 | p<0.001 | 5 | 95.36 [93.87,96.84] | 4 | 87.81 [80.51,95.11] | 1 | 92.0 [88.93,95.06] | 1 | 72.00 [68.79,75.20] | 1 | 65.30 [61.98,68.61] | |
| Girls used both soap and water | 17 | 50.69 [41.53,59.84] | 98.9 | p<0.001 | 8 | 59.54 [48.86,70.23] | 6 | 40.80 [30.25,51.35] | 1 | 41.90 [36.32,47.47] | 1 | 29.50 [26.24,32.75] | 1 | 69.50 [66.29,72.71] | |
| Girls use only water | 11 | 45.85 [39.47,52.23] | 95.9 | p<0.001 | 4 | 43.15 [38.07,48.24] | 5 | 43.91[33.01, 54.80] | 1 | 49.50 [43.85,55.15] | 1 | 62.50 [39.48,52.23] | - | - | |
| Frequency of washing (at least two times/day) | 7 | 41.67 [27.12,56.23] | 98.8 | p<0.001 | 3 | 51.84 [17.74,85.93] | 3 | 35.75 [32.16,39.35] | 1 | 27.90 [22.83,32.97] | - | - | - | - | |
| *Disposal of absorbent material* | | | | | | | | | | | | | | | |
| Throw in the latrine | 14 | 62.18 [52.87,71.49] | 98.7 | p<0.001 | 5 | 65.38 [40.99,89.77] | 7 | 62.44 [53.57,71.31] | 1 | 54.20 [48.57,59.83] | 1 | 52.40 [49.40,55.39] | - | - | |
| Open field | 13 | 15.85 [9.23,22.46] | 99.3 | p<0.001 | 5 | 17.22 [3.63,30.81] | 6 | 10.28 [5.22,15.34] | 1 | 11.00 [7.46,14.53] | 1 | 46.20 [43.21,49.19] | - | - | |
| Put in the dust bin | 14 | 26.96 [16.52,37.41] | 99.5 | p<0.001 | 6 | 24.32 [5.76,42.88] | 6 | 25.72 [11.02,40.40] | 1 | 32.60 [27.30,37.89] | - | - | 1 | 44.70 [41.23,48.16] | |
| Disposes pads by wrapping with paper | 3 | 27.35 [18.55,36.14] | 95.8 | p<0.001 | 2 | 26.04 [11.83,40.24] | - | - | - | - | - | - | 1 | 30.00 [26.81,33.19] | |
| Burn | 1 | 11.80 [8.95,14.65] | - | - | 1 | 11.80 [8.91,14.65] | - | - | - | - | - | - | - | - | |
| Burying | 1 | 6.60 [3.66,9.54] | - | - | 1 | 6.60[3.66,9.54] | - | - | - | - | - | - | - | - | |
| *Bath during menstruation* | | | | | | | | | | | | | | | |
| Daily & ≥ 2 times/day | 13 | 52.59 [40.45,64.73] | 99.2 | p<0.001 | 6 | 51.34 [44.01,58.68] | 4 | 50.61 [18.23,82.98] | 1 | 50.20 [45.66,54.74] | 1 | 66.6 [63.03,70.17] | 1 | 56.40 [51.51,61.29] | |
| *School absenteeism during menstruation* | | | | | | | | | | | | | | | |
| Left school (at least one day and above) | 7 | 32.03 [22.65,41.40] | 98.2 | p<0.001 | 3 | 28.38 [19.35,37.41] | 2 | 37.89 [5.45,70.33] | - | - | 2 | 31.72 [7.12,56.32] | - | - | |

# number of included studies.

primary studies, but none of these variables was found to be a statistically significant source of heterogeneity (**Table 5**). The results of meta-regression analysis also showed no significant relationship between safe MHM and sampled size (p = 0.853) (**Fig 5**) or over time (p = 0.952) (**Fig 6**).

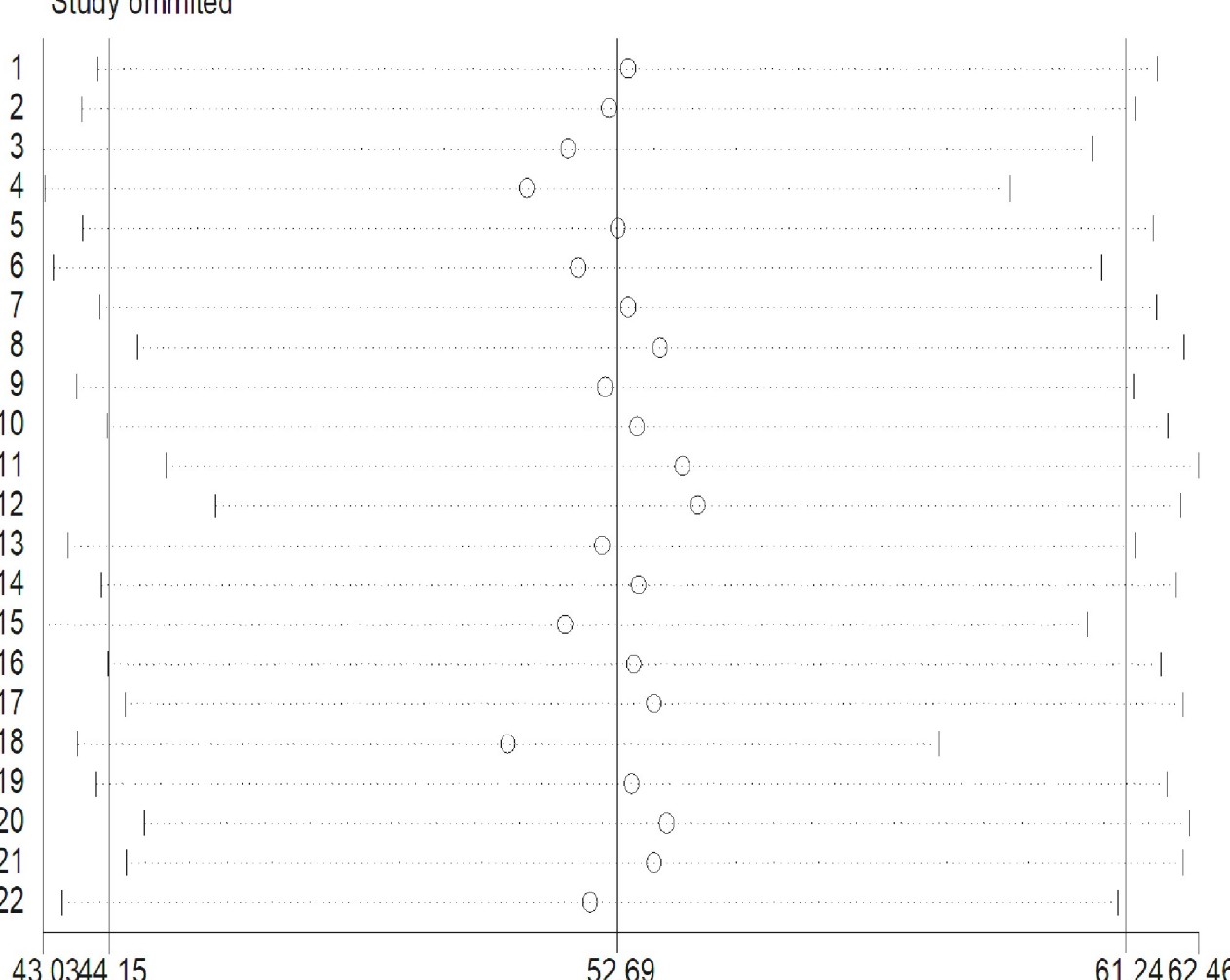

**Fig 4. Sensitivity analysis for estimates on safe menstrual hygiene management proportion among adolescent girls in Ethiopia (number of estimates = 22).**

### Publication bias

In this meta-analysis, possible publication bias was also visualized through funnel plots and using Begg's test. Asymmetrical large inverted funnel resembled the absence of publication biases (Fig 7). Also, the probability of publication biases was tested using Egger's weighted regression and Begg's rank correlation test. In the current review, Begg's and Egger's tests were

**Table 5. Related factors with the heterogeneity of safe MHM proportion in the current meta-analysis (based on univariate meta-regression).**

| Variables | Coefficient | p-value |
|---|---|---|
| Years of publication | -0.261 | 0.805 |
| Sample size | -0.024 | 0.118 |
| Quality Score | 3.212 | 0.490 |

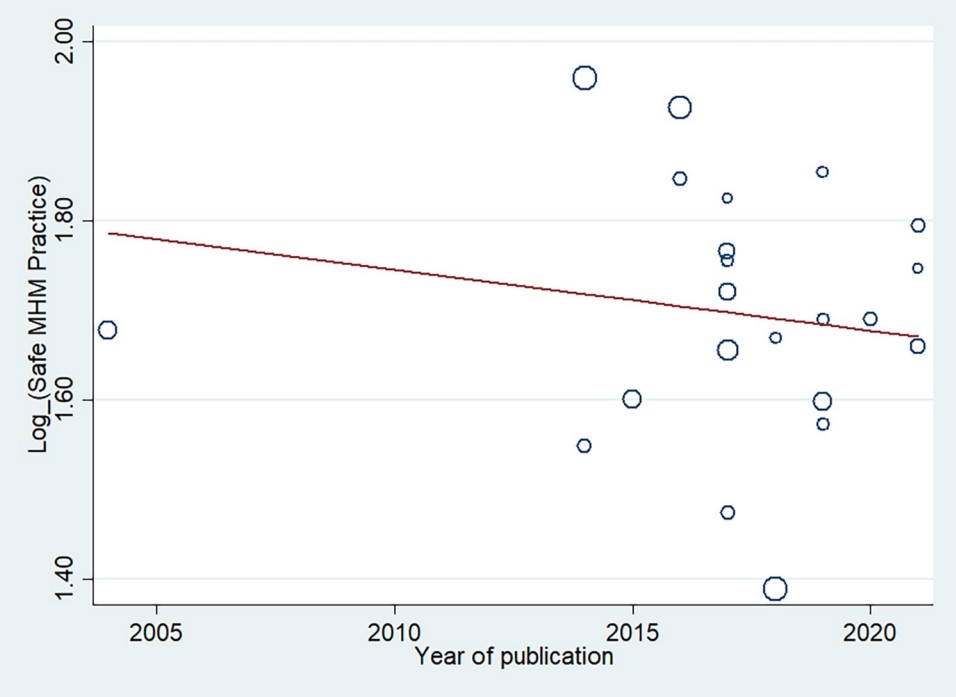

**Fig 5. Reduction in proportion of safe MHM during 2004–2021 according to meta-regression.** This figure shows a meta-regression analysis of safe MHM proportion based on selected studies' publication years. The vertical axis represents the log proportion of safe MHM, and the horizontal axis represents the selected studies' publication year. Each circle demonstrates one selected study, and the size of each circle corresponds to the weight assigned to each study. The slope of the regression line indicates an increase or decrease of the study effect using REML estimation. Given the slope of the regression line is descending in this figure, it can be inferred that as the studies' publication year has been increased, the proportion of safe MHM has been decreased. However, this association was not statistically significant (p = 0.952).

not statistically significant for the estimated proportion of safe MHM among adolescent girls in Ethiopia with a p-value of 0.310 and 0.489, respectively. **Fig 8** also showed Egger's regression test of publication bias of studies.

## Evidence from the reviewed studies

As shown in **Table 2**, the most consistent factors associated with safe MHM in Ethiopia included (i) socio-demographic and economic factors such as maternal and paternal educational and occupational status, place of residence, monthly income, and access to media sources like television (ii) adolescent girls characteristics such as age, school grade, and earning pocket money, and receive advice on menstruation (iii) access to WASH facilities, (iv) awareness/ information about menstruation before menarche and knowledge about menstruation.

The review showed that higher socioeconomic status [7], higher maternal education [20, 24, 33, 36, 37], higher paternal education [13], maternal employment [20], access to water facility [12, 38, 41], access to toilet facility [12, 19], adolescent girls who earn pocket money from their families [20, 26, 32], and girls who reported that they used media (electronic/books) as common source of information about menses [13], and schoolgirls with good knowledge about menstruation [19, 21, 24, 25, 36–38, 40] were associated with higher odds of safe MHM.

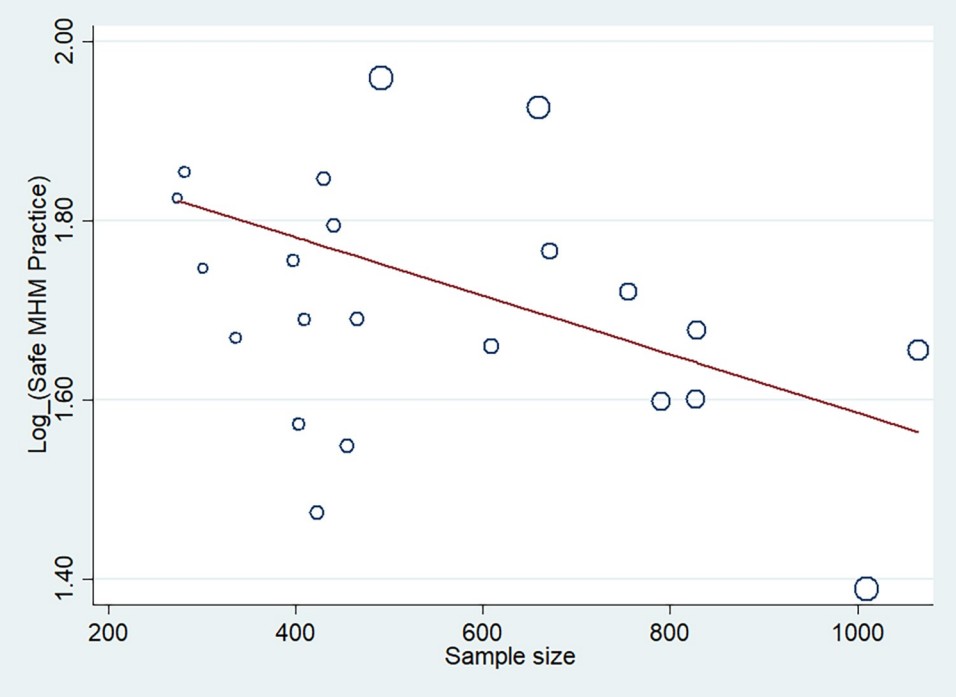

**Fig 6. Proportion of safe MHM among adolescent girls based on the sample size of selected studies by meta-regression analysis.** Circles show the weight of the included studies. The figure indicates the association between the safe MHM and sample size. The proportion of safe MHM was reduced with a rise in sample size. However, this association was not statistically significant (p = 0.853).

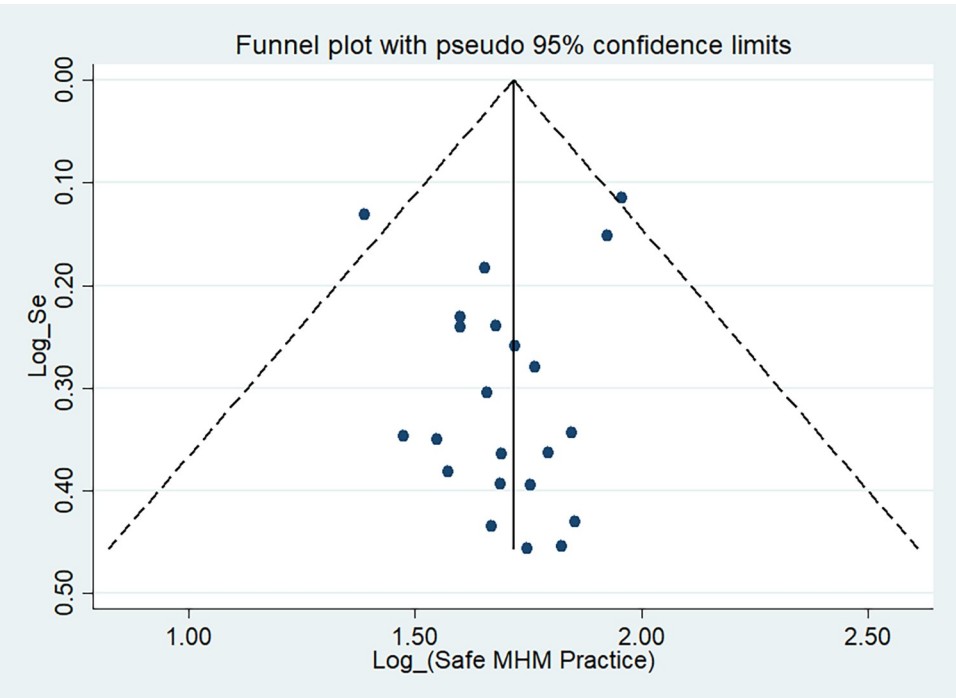

**Fig 7. Funnel plot showing publication bias of proportion of MHM studies among adolescent girls in Ethiopia.**

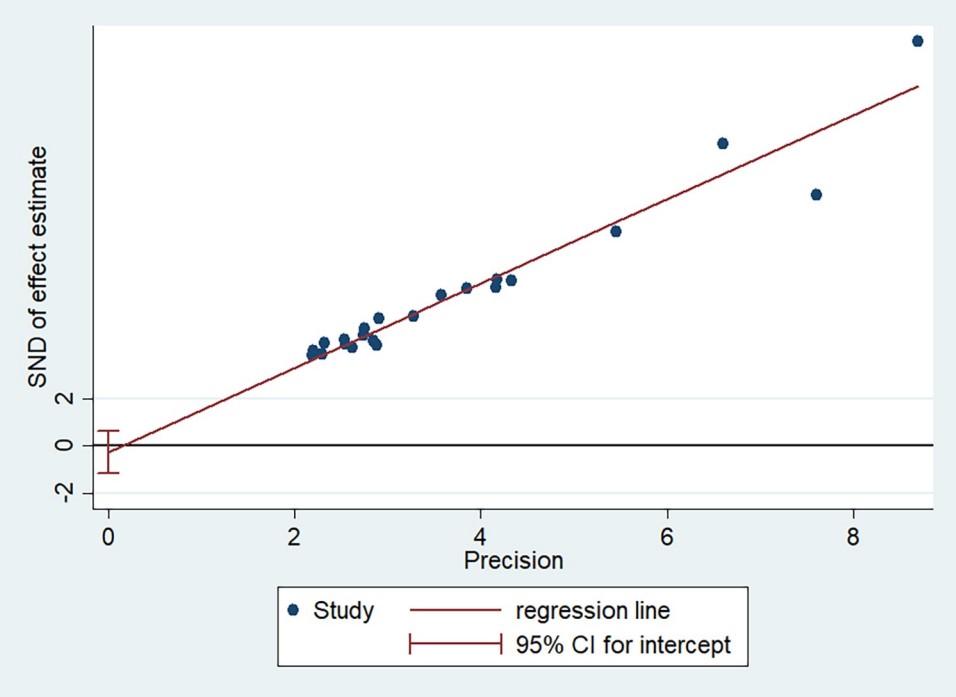

**Fig 8. Publication bias.** Dots show included studies, the horizontal axis represents accuracy, and the vertical axis represents the standardized effect. The line shown in this figure is a regression line related to Egger's regression test. It shows that whether this line cut the vertical axis at the point near zero or not. If this line distance from zero, it indicates a bias in publishing the results. If there is no publication bias, it is expected that this line passes from origin to a point near the origin. Since intercept (width from origin) is close to 0 in this figure, we conclude that there is no publication bias. Because zero is included in the confidence interval. Therefore, it could be concluded that the publication bias is not significant.

## Discussion

Menstrual Hygiene Management (MHM) is vital to the empowerment of adolescent school girls. It is about more than just access to low-cost sanitary menstrual pads and suitable toilets–though those are imperative. It is also about ensuring girls live in an environment that values and supports their ability to manage their menstruation with dignity [22]. Poor MHM not only puts girls at risk of infection, but it would affect girl's education, confidence, and self-esteem in a major way. In Ethiopia, as in many developing countries, menstrual hygiene management among adolescent school girls remains poorly discussed so far. The current comprehensive systematic review and meta-analyses have dealt with the proportion of safe MHM and its associated factors in Ethiopia.

The review has included 22 primary studies investigating MHM among adolescent girls. The review founds that only half of the adolescent girls in Ethiopia had safe MHM practice, a significant number of girls use homemade cloth or disposable piece of rags (53.03%) for their MHM, inappropriate disposal was common (62.2%), and one in four girls missing one or more school days during menstruation (32.03%). We also found that adolescent girls who live in an urban area, those who earn pocket money from their parents and have good knowledge on menstrual hygiene were positively associated with safe MHM.

The current meta-analysis showed that almost one in two adolescent girls had safe MHM in Ethiopia, and 64.6% use commercial menstrual absorbents. These findings were higher than a systematic review conducted by van Eijk et al. in India [5], which reported 45% of girls-only

use commercial menstrual absorbents. Although it is favorable to use commercial sanitary pads, use of commercial absorbents varies across Ethiopian regions 87.6% (in Addis Ababa, the capital city) to SNNP region 42.4%. Different studies also reported variation across urban and rural dwellers, particularly in rural areas low commercial menstrual absorbent was frequently reported in Asia [5, 42] and African countries [43, 44]. As the current cost of sanitary pads is too high for many girls, they remain dependent on reusable cloth pads that they wash and reuse [45, 46]. For instance, one study from rural Kenya showed that two-thirds of girls and young women aged 13–29 using sanitary pads reported receiving them from sexual partners. There was a higher likelihood of receiving pads if respondents had more than one sexual partner, placing girls at increased risk of HIV or unwanted pregnancy. The prevalence of sex for money to purchase pads was concentrated among 15-year-old girls, with a six-fold higher likelihood of engaging in this practice than older respondents [47]. As cloths are traditionally used to absorb menstrual flow, and they are cheaper, a significant number of adolescent girls use homemade close and disposal rage to absorb menstrual flows in this review, which was comparable to studies conducted elsewhere [45, 48]. In this review, low use of tissue paper was reported as absorbents by girls, which is much lower than a study report from Uganda, 37.1% [46].

The disposal of menstrual absorbents, particularly commercial pads, is of great concern because of their high content of non-biodegradable components [5]. As the conventional sanitary pads available in the market are loaded with plastics, sanitary products usually made out of plastic take over 500 years to fully degrade [42]. In India, every year, about 12.3 billion sanitary napkins, the majority of which are not biodegradable or compostable, were disposed of, and 23 percent throw the waste in open spaces like drains, rivers, wells, lakes, and roadside [49]. In this review, remarkably two-third (62.2%) of used menstrual absorbents end up in the latrine, and the practice was comparable in all regions of Ethiopia. In addition, 15.8% of girls threw the absorbent in an open field. In Ethiopia, the safe disposal of menstrual absorbents has become a growing problem across Ethiopian regions. However, the problem receives little attention. There is also no standardized method for disposing of such pads; in most cases, they are thrown with regular waste, thrown in the open, buried, or burned openly. Our pooled estimate of burying (6.6%) and burning (11.8%) of the used pad was lower than that in a systematic review report from India, where 25% and 17% of adolescent girls across India reported they either burying or burning used menstrual absorbents, respectively [5].

In the current review, 50.7% of schoolgirls reported they used both soap and water for genital cleaning, and 45.8% reported washing their genitalia using water only. In Ethiopia, nearly 39 million people in rural areas do not have access to a safe water supply. As the majority of them were adolescent girls, maintaining the use of clean water and soap for genital cleaning for girls remains a significant problem. Furthermore, the WASH infrastructure in schools in Ethiopia is currently in a very poor state, with over half of girls (56%) reporting there is never access to water in schools and over three quarters (77%) of communal and school toilets observed had faecal matter present in the facilities [50]. Because of these and many cultural problems, adolescent girls face a major barrier to ensuring appropriate MHM, which puts them at risk of infection. It was pointed that the cleanliness of the genital area and access to safe menstrual hygiene products can reduce the incidents of infection up to 97%. Changing sanitary napkins every 4 hours and washing hands whenever sanitary napkins are changed is a significant step towards ensuring good hygiene during periods [51]. A study by Das et al. found that washing (bath or vaginal wash) with water only as compared with water and soap during menstruation was associated with symptomatic cases of urogenital infection (OR: 2.4, 95% CI:1.01–5.7, p = 0.045) [16].

The poor state of MHM significantly disrupts the academic achievement of adolescent girls. In this review, one in four girls missed school one or more school days during menstruation. Similarly, girls from resource-poor countries worldwide attribute frequent school absences to difficulties managing their menses [4, 5, 7, 31, 52]. For instance, nearly two-thirds of adolescent girls miss school in Uganda at least once per month because of menstruation [46]. In many cases, school absenteeism appears to be closely associated with lack of privacy, gender-specific WASH facilities, and limited availability of water and sanitation facilities at schools [4, 52, 53]. Moreover, during menstruation and fear of sudden menstrual blood leakage, as many girls in developing countries did not use proper sanitary napkins, they were also the contributing factor for students' dropout, and a key reason for attending class attentively possibly leads to school absenteeism [4, 8]. For this reason, a gender-specific and gender-sensitive school environment that supports adolescent girl's menstrual hygiene rights is important for school girls to engage in class and stay in school.

## Limitations

This review has several limitations. First, all included studies were cross-sectional, so causality cannot be inferred. Second, because the included studies used self-administered questionnaires or interviews, social desirability bias was likely. Third, many of the included studies use different definitions for good MHM, despite the fact that they have commonly included items. For this reason, we carefully extract each outcome definition from the included studies along with their prevalence. Fourth, there was high heterogeneity between included studies as indicated by the $I^2$ statistic, which was frequently over 90%. This may be explained by the different methodology of the studies and the different settings where the studies were conducted. To overcome this, we performed sub-group analysis, sensitivity analysis, and meta-regression to explore factors that affected the outcomes. Six, the inclusion of unpublished studies may be a limitation of this study because some unpublished manuscripts may not eventually meet with a quality peer reviewed article. Seven, no study was obtained from some Ethiopian regions, such as Afar, Benshangul-Gumuz, Tigray, Somali, Gagmbela regions, and Dire Dawa city administrations affect the representatives of our finding. Seven, our review was limited by the information available; not all included studies focused solely on MHM.

## Conclusions

This study revealed that only half of the adolescent girls in Ethiopia had safe MHM practices. For menstrual absorbents, more than half of adolescent girls use homemade cloth. Disposal of absorbent material into the latrine was the most common practice in Ethiopia. The study also revealed that one in four girls reported missing one or more school days during menstruation. To ensure that girls in Ethiopia can manage menstruation hygienically and with dignity; strong gender-specific WASH facilities along with awareness creation activities at every level are needed. Given the numerous challenges that adolescent girls face in the country, it is evident that promoting MHM is more than just a sanitation issue; it is also an important step toward protecting girls' dignity, bodily integrity, and overall life opportunities. Above all, schoolgirls require effective, safe, and free/affordable menstrual hygiene products to improve menstrual hygiene practice and support girls school attendance.

## Supporting information

**S1 File. Preferred Reporting Items for Systematic Reviews and Meta-Analyses (PRISMA).** (DOC)

**S2 File. Examples of searching strategy.**
(DOCX)

**S3 File. Individual quality assessment of 22 articles included in the review of the status of menstrual hygiene management among adolescent girls in Ethiopia, studies published between 2004 and 2021.**
(DOCX)

**S4 File. Sensitivity analysis for estimates on safe MHM proportion among adolescent girls in Ethiopia.**
(DOCX)

## Acknowledgments

The authors would like to thank Madda Walabu University Goba Referral Hospital Public Health Department staff for providing their unreserved support.

## Author Contributions

**Conceptualization:** Biniyam Sahiledengle.

**Data curation:** Biniyam Sahiledengle, Daniel Atlaw.

**Formal analysis:** Biniyam Sahiledengle, Daniel Atlaw.

**Investigation:** Biniyam Sahiledengle, Daniel Atlaw.

**Methodology:** Biniyam Sahiledengle.

**Project administration:** Biniyam Sahiledengle.

**Resources:** Biniyam Sahiledengle.

**Software:** Biniyam Sahiledengle.

**Supervision:** Abera Kumie, Kingsley Emwinyore Agho.

**Validation:** Abera Kumie, Yohannes Tekalegn, Demelash Woldeyohannes, Kingsley Emwinyore Agho.

**Visualization:** Biniyam Sahiledengle, Abera Kumie, Yohannes Tekalegn, Demelash Woldeyohannes, Kingsley Emwinyore Agho.

**Writing – original draft:** Biniyam Sahiledengle.

**Writing – review & editing:** Biniyam Sahiledengle, Daniel Atlaw, Abera Kumie, Yohannes Tekalegn, Demelash Woldeyohannes, Kingsley Emwinyore Agho.

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
