## [Decision Letter · Decision Letter 0]

24 Nov 2021

PONE-D-21-22639Menstrual Hygiene Practice among Adolescent Girls in Ethiopia: a systematic review and meta-analysisPLOS ONE

Dear Dr. Sahiledengle,

Thank you for submitting your manuscript to PLOS ONE. After careful consideration, we feel that it has merit but does not fully meet PLOS ONE’s publication criteria as it currently stands. Therefore, we invite you to submit a revised version of the manuscript that addresses the points raised during the review process.

We look forward to receiving your revised manuscript.

Kind regards,

Balasubramani Ravindran, Ph.D

Academic Editor

PLOS ONE

Journal Requirements:

Reviewers' comments:

Reviewer's Responses to Questions

**Comments to the Author**

1. Is the manuscript technically sound, and do the data support the conclusions?

Reviewer #1: Yes

Reviewer #2: Yes

Reviewer #3: Yes

2. Has the statistical analysis been performed appropriately and rigorously? 

Reviewer #1: Yes

Reviewer #2: Yes

Reviewer #3: No

3. Have the authors made all data underlying the findings in their manuscript fully available?

Reviewer #1: Yes

Reviewer #2: Yes

Reviewer #3: Yes

4. Is the manuscript presented in an intelligible fashion and written in standard English?

Reviewer #1: Yes

Reviewer #2: Yes

Reviewer #3: Yes

5. Review Comments to the Author

Reviewer #1: This study is not a primary scientific research, it is a systematic review and meta-analysis. Therefore, it does not satisfy the first publishing criteria of PLOS ONE. However, it should be noted that the manuscript is technically sound and well written. The manuscript substantially met remaining requirements for publication in PLOS ONE.

Reviewer #2: Thank you. The article is well written and relevant as menstrual hygiene is vital to the empowerment and well-being of women and girls worldwide. Safe menstrual hygiene practice is especially important for adolescent girls as it affects girls’ participation and psychological well-being while in classes. Below are some minor comments:

Abstract:

Line 35: please write out “MHM” before using abbreviation

Introduction:

Line 95-99: wordy and the meaning is not clear. Suggest rephrasing of this sentence “Despite, a school WASH initiative includes MHM as a major focus, as part of its overall objective of improving the school enrolment and attendance of girls was implemented by the Federal Ministry of Health (FMoH) of Ethiopia, still, poor access to dignified, clean, and functional WASH facilities in schools in Ethiopia remains a major setback for proper MHM by adolescent girls.”

Method:

Line 136-138: These 2 sentences discussing exclusion criteria should be moved to Exclusion criteria.

Line 145: as mentioned in point above, the exclusion criteria should include “Systematic reviews, commentaries, letters to editors, short communications, qualitative studies were excluded. Also, articles that were not fully accessible after two-email contact with the primary/corresponding author were excluded.”

Result:

Detailed, clear and well-written.

Line 274-276: Please rephrase the sentence and clarify if the 41.67% of schoolgirls had access to water only OR water and soap OR either, “However, 50.69% (95%CI: 41.53, 59.84, I2 98.9%) of schoolgirls reported they used both soap and water for genital cleaning, and 41.67% (95%CI: 27.12-56.23, I2 98.8%) reported wash their genitalia daily or two-time a day during menstruation”

Discussion:

Line 422: The message is not clear, please rephrase the sentence “However, the problem does not give due attention as more adolescent girls turn to commercial pads (65%), with the potential generation of non-biodegradable wastes.”

Conclusion:

Well written with summary of the research question and recommendation.

General:

Few typographical errors were identified throughout the manuscript. Please address typo/grammatical errors highlighted in the pdf file.

Reviewer #3: This study is important and would be very useful for policy makers and other stakeholders to develop interventions that will be appropriate for improving safe menstrual hygiene management especially in developing countries. These are my reviews

1. Introduction

The introduction is clear. The objectives are well spelt out

2. Methods

line 145: Exclusion criteria should also include population that are not within the adolescent age bracket

line 164 should be a continuous statement under primary outcome, it need not to be on a fresh paragraph

Line 165 should be on a new paragraph stating the secondary outcome clearly

3. Results

The secondary outcome stated under methods which is the safe menstrual practice and association between sociodemgraphic and behavioural factors is not clearly evaluated. Table 5, line 319. This makes the study needing a major revision

4. Discussion

The inclusion of unpublished studies may be a limitation to this study as some of the unpublished manuscript might not eventually meet up with quality peer reviewed article.

6. PLOS authors have the option to publish the peer review history of their article (what does this mean?). If published, this will include your full peer review and any attached files.

Reviewer #1: No

Reviewer #2: No

Reviewer #3: No

---

## [Author Response · Author response to Decision Letter 0]

13 Dec 2021

Response to Reviewers

Dear Editor,

Receiving such valuable comments and suggestions to improve the current manuscript is a prestigious opportunity for us. The comments and suggestions taught us a lot. We improved our manuscript based on the reviewers' suggestions, and we used "Green Text Highlight Color" for all affected revisions and corrections. In the attached word document, we also included a point-by-point response to the editor's and reviewers' comments.

Thank you. 

Response to Reviewer #1: 

Reviewer #1: This study is not a primary scientific research, it is a systematic review and meta-analysis. Therefore, it does not satisfy the first publishing criteria of PLOS ONE. However, it should be noted that the manuscript is technically sound and well written. The manuscript substantially met remaining requirements for publication in PLOS ONE.

Response: 

Receiving such valuable comments and suggestions is a prestigious opportunity for us. Thank you for your valuable time and constructive feedback. 

Response to Reviewer #2:

Thank you for this wonderful opportunity, respected reviewer #3. The comments and suggestions taught us a lot. We have improved our manuscript in response to your wise advice. For all affected revisions and corrections, we used "Green Text Highlight Color." Please refer to the updated manuscript. Thank you very much.

Reviewer #2: Thank you. The article is well written and relevant as menstrual hygiene is vital to the empowerment and well-being of women and girls worldwide. Safe menstrual hygiene practice is especially important for adolescent girls as it affects girls’ participation and psychological well-being while in classes. Below are some minor comments:

Abstract:

Line 35: please write out “MHM” before using abbreviation

Response:

Thank you, our respected reviewer #2. As per your wise advice we write out MHM accordingly. Please see the revised manuscript abstract section. Thank you.

Introduction:

Line 95-99: wordy and the meaning is not clear. Suggest rephrasing of this sentence “Despite, a school WASH initiative includes MHM as a major focus, as part of its overall objective of improving the school enrolment and attendance of girls was implemented by the Federal Ministry of Health (FMoH) of Ethiopia, still, poor access to dignified, clean, and functional WASH facilities in schools in Ethiopia remains a major setback for proper MHM by adolescent girls.”

Response:

Thank you, our respected reviewer #2. We apologize for this confusing statement. As per your wise advice we rephrased the stated introduction section accordingly. Please refer to the updated manuscript. Thank you. 

Method:

Line 136-138: These 2 sentences discussing exclusion criteria should be moved to Exclusion criteria.

Response:

Thank you, our respected reviewer. As per your wise advice we removed the two sentences in the exclusion criteria. Thank you.

Line 145: as mentioned in point above, the exclusion criteria should include “Systematic reviews, commentaries, letters to editors, short communications, qualitative studies were excluded. Also, articles that were not fully accessible after two-email contact with the primary/corresponding author were excluded.”

Response:

Thank you, our respected reviewer. As per your wise advice corrected accordingly. Thank you with all respect.

Result:

Detailed, clear and well-written.

Line 274-276: Please rephrase the sentence and clarify if the 41.67% of schoolgirls had access to water only OR water and soap OR either, “However, 50.69% (95%CI: 41.53, 59.84, I2 98.9%) of schoolgirls reported they used both soap and water for genital cleaning, and 41.67% (95%CI: 27.12-56.23, I2 98.8%) reported wash their genitalia daily or two-time a day during menstruation”

Response:

Thank you, our respected reviewer. We apologize for the confusing statement. As per your wise suggestion we clarify and rewire accordingly. Please refer to the updated manuscript. Thank you with all respect.

Discussion:

Line 422: The message is not clear, please rephrase the sentence “However, the problem does not give due attention as more adolescent girls turn to commercial pads (65%), with the potential generation of non-biodegradable wastes.”

Response:

Thank you, our respected reviewer. We apologize for the confusing rephrase. As per your wise suggestion we clarify and rewire accordingly. Please refer to the updated manuscript. Thank you with all respect.

Conclusion:

Well written with summary of the research question and recommendation.

Response:

Thank you, our esteemed reviewer. 

General:

Few typographical errors were identified throughout the manuscript. Please address typo/grammatical errors highlighted in the pdf file.

Response: Thank you for your time, our respected Reviewer #2. We apologize for the typographical and grammatical errors. In the revised manuscript we correct and rewired in accordance with your wise advice. Please refer to the updated manuscript. Thank you.

Response to Reviewer #3:

Thank you for this wonderful opportunity, respected reviewer #3. The comments and suggestions taught us a lot. We have improved our manuscript in response to your wise advice. For all affected revisions and corrections, we used "Green Text Highlight Color." Please refer to the updated manuscript. Thank you very much.

Reviewer #3: This study is important and would be very useful for policy makers and other stakeholders to develop interventions that will be appropriate for improving safe menstrual hygiene management especially in developing countries. These are my reviews

1. Introduction

The introduction is clear. The objectives are well spelt out

Response

Thank you our respected reviewer. 

2. Methods

line 145: Exclusion criteria should also include population that are not within the adolescent age bracket

Response 

Thank you our respected reviewer. As per your wise advice we include population that are not within the adolescent age in the exclusion criteria. Please refer to the updated manuscript. Thank you very much.

line 164 should be a continuous statement under primary outcome, it need not to be on a fresh paragraph

Line 165 should be on a new paragraph stating the secondary outcome clearly

Response 

Thank you our respected reviewer. As per your wise advice we include revised this section accordingly. Please refer to the updated manuscript. Thank you very much.

3. Results

The secondary outcome stated under methods which is the safe menstrual practice and association between sociodemgraphic and behavioural factors is not clearly evaluated. Table 5, line 319. This makes the study needing a major revision

Response

Thank you for your time, Mr. Reviewer #3. We sincerely apologize for the error we made while writing our draft manuscript. As stated in the introduction, we did not have a secondary objective in this study. “This systematic review and meta-analysis aim to estimate: (1), the pooled proportion of safe MHM, and (2) the pooled estimate for the type of absorbent used by adolescent girls, disposal practice of absorbents, hygiene during menstruation, bathing during menstruation, and school absenteeism were measured using available studies.”

We removed line 319 “The second outcome of the study was to determine the association between safe menstrual hygiene practice and socio-demographic and behavioral factors.”.

Regarding Table 5, we used a univariate meta-regression to identify potential sources of heterogeneity across primary studies. Please refer to the updated manuscript. Thank you very 

4. Discussion

The inclusion of unpublished studies may be a limitation to this study as some of the unpublished manuscript might not eventually meet up with quality peer reviewed article.

Response

Thank you for your time, our respected Reviewer #3. According to your wise advice, we describe the inclusion of unpublished studies as one of the study's limitations. Please see the revised manuscript. Thank you kindly.

---

## [Decision Letter · Decision Letter 1]

21 Dec 2021

Menstrual Hygiene Practice among Adolescent Girls in Ethiopia: a systematic review and meta-analysis

PONE-D-21-22639R1

Dear Dr. Biniyam Sahiledengle,

We’re pleased to inform you that your manuscript has been judged scientifically suitable for publication and will be formally accepted for publication once it meets all outstanding technical requirements.

Kind regards,

Balasubramani Ravindran, Ph.D

Academic Editor

PLOS ONE

Reviewers' comments:

Reviewer's Responses to Questions

**Comments to the Author**

1. If the authors have adequately addressed your comments raised in a previous round of review and you feel that this manuscript is now acceptable for publication, you may indicate that here to bypass the “Comments to the Author” section, enter your conflict of interest statement in the “Confidential to Editor” section, and submit your "Accept" recommendation.

Reviewer #1: All comments have been addressed

Reviewer #2: All comments have been addressed

2. Is the manuscript technically sound, and do the data support the conclusions?

Reviewer #1: Yes

Reviewer #2: Yes

3. Has the statistical analysis been performed appropriately and rigorously? 

Reviewer #1: Yes

Reviewer #2: Yes

4. Have the authors made all data underlying the findings in their manuscript fully available?

Reviewer #1: Yes

Reviewer #2: Yes

5. Is the manuscript presented in an intelligible fashion and written in standard English?

Reviewer #1: Yes

Reviewer #2: Yes

6. Review Comments to the Author

Reviewer #1: Besides making the suggested corrections by the reviewers, the author has responded satisfactorily to the reviewers' comments. This has significantly improved the overall quality of the manuscript. Therefore, I will recommend that the manuscript be accepted for publication.

Reviewer #2: Thank you for revising the manuscript and responding to all comments. There is one sentence that is still unclear even after rephrasing.

Discussion: Line 420; “However, the issue does not pay enough attention to raising awareness about proper disposal of such pads”, unclear/redundant statement.

7. PLOS authors have the option to publish the peer review history of their article (what does this mean?). If published, this will include your full peer review and any attached files.

Reviewer #1: No

Reviewer #2: No

---

## [Editor Report · Acceptance letter]

26 Dec 2021

PONE-D-21-22639R1 

Menstrual Hygiene Practice among Adolescent Girls in Ethiopia: a systematic review and meta-analysis 

Dear Dr. Sahiledengle:

I'm pleased to inform you that your manuscript has been deemed suitable for publication in PLOS ONE. Congratulations! Your manuscript is now with our production department. 

Kind regards, 

on behalf of

Dr. Balasubramani Ravindran 

Academic Editor

PLOS ONE